# Measurement invariance of the Center for Epidemiological Studies-Depression scale and associations with genetic risk in older adults

Toni T. Saari[1,2]*, Maarit Piirtola[1,3], Aino Aaltonen[1], Teemu Palviainen[1], Anni Varjonen[1], Valtteri Julkunen[2], Juha O. Rinne[4,5], Jaakko Kaprio[1], Eero Vuoksimaa[1]

**1** Institute for Molecular Medicine Finland (FIMM), HiLIFE, University of Helsinki, Helsinki, Finland,
**2** Department of Neurology, University of Eastern Finland, Kuopio, Finland, **3** UKK Institute for Health Promotion Research, Tampere, Finland, **4** Turku PET Centre, Turku University Hospital, Turku, Finland,
**5** University of Turku, Turku, Finland

* toni.saari@helsinki.fi

**Data Availability Statement:** Data cannot be shared publicly because of the consent given by study participants and the high potential for

## Abstract

### Background

As populations are aging, it needs to be ensured that valid depression rating scales are available across old adulthood. Center for Epidemiological Studies-Depression scale (CES-D) is a common depression rating scale, however, few studies have assessed its validity in individuals with age over 90 and/or cognitive impairment. We examined the factor structures of 20-, 15-, and 8-item CES-D scales, their measurement invariance for age and cognition, and associations with genetic risk of depression.

### Methods

Participants were from a population-based older Finnish Twin Cohort study including 71–79-year-olds from the MEMTWIN II (n = 1034 for exploratory and n = 664 for confirmatory factor analyses) and 90+ year-olds from the NONAGINTA (n = 134, confirmatory factor analyses) sub-studies. Associations of polygenic risk score of major depressive disorder (MDD-PRS) with CES-D scales were examined in MEMTWIN II.

### Results

Exploratory factor analyses (n = 1034) suggested four- (CES-D 20) and three-factor (CES-D 8) structures and these models fit well in confirmatory analyses (n = 664). Unidimensional models had good (CES-D 15 & 20) or fair fit (CES-D 8). Results supported scalar invariance of all CES-D versions for age and cognitive status. Higher MDD-PRS was associated with more depressive symptoms in different CES-D versions.

### Conclusions

Different CES-D versions are adequate for measuring depressive symptoms across age groups and cognitive spectrum in old age. Genetic risk of depression predicts depressive symptoms even in old age.

identification of participants from the data. The data can be requested through the Institute for Molecular Medicine Finland (FIMM) Data Access Committee (DAC) for authorized researchers who have ethical approval and an approved study plan. For details, please contact the FIMM DAC (fimm-dac@helsinki.fi).

**Funding:** Data collection and analyses in the Finnish Twin Cohort have been supported by the Academy of Finland (grants to JK: 312073, 336823, 352792). MEMTWIN II data collection was supported by the Sigrid Juselius foundation, The Academy of Finland (grants 133193 and 310962) and Finnish Governmental Research Funding (VTR). NONAGINTA study was funded by the Academy of Finland grants (320109 and 345988 to EV). EV was supported by the Academy Research Fellow grant 314639 and the Sigrid Jusélius Foundation. Open access funded by Helsinki University Library.

**Competing interests:** JK has received support from Sigrid Jusélius Foundation, the European Union and the National Institutes of Health. JOR has consulted for Clinical Research Services Turku and acted as a member on the data monitoring committee for Lundbeck.

## Introduction

Late-life depression is a growing burden to patients, health care systems, and societies [1]. In a recent meta-analysis of older adults, depressive symptoms were more prevalent in over 75-year-old individuals compared to under 75-year-old individuals [2], likely reflecting increased health and psychosocial challenges in old age [3, 4]. However, the stability of depression prevalence across the adult life has also been stated [5]. Prevalence estimates of depression in older adults vary from 7% to nearly 40% [2, 6], and similar figures have been reported in older adults with cognitive impairment [7, 8].

The Center for Epidemiological Studies Depression scale (CES-D) is a self-report instrument for depressive symptoms with psychometric support in diverse populations, including older adults [9–13]. In the original study, Radloff [9] suggested a four-domain structure of the CES-D which has received meta-analytic support [14]. To reduce the length, various short forms of the CES-D have been developed. An 8-item version of the CES-D has received psychometric support in studies of older adults [3, 15–17], but the number of factors in these studies has varied. In terms of diagnostic properties, the CES-D 8 has good sensitivity and specificity when comparing it to the long version cut-off for clinically significant depressive symptoms [13].

Despite the validity evidence for the CES-D as a measure of depressive symptoms in older adults, few studies have examined whether the CES-D is invariant for age and cognition. Comparison of depressive symptoms across age and cognitive groups is only meaningful if measurement invariance has been established [3, 18]. In older adults, the CES-D has demonstrated measurement invariance for age comparing 20–54 to over 55-year-olds [19] and 50–64 to over 65-year-olds [3]. In over 65-year-olds, the CES-D 8 has demonstrated temporal measurement invariance, meaning the factor structure has stayed the same across multiple follow-ups of the same persons [15]. However, there is no information about measurement invariance comparing older adults and those who are 90 years or older, the fastest growing population segment in many countries. Thus, it remains unclear whether the CES-D has a similar factor structure in the oldest-old compared to younger old adults.

Relatively few studies have investigated whether the CES-D is invariant for cognitive status. In one study, diagnostic accuracy of the CES-D did not markedly differ between individuals with and without dementia [20]. Another study using differential item functioning found small differences between cognitively healthy and cognitively impaired, but these differences were not sufficiently large to suggest that the CES-D functions differently across cognitive states [21]. Considering that the proportion and number of older adults in populations around the world are increasing [22] and the risk of dementia increases with age [23], it must be ensured that suitable depression rating scales are available for older adults across age and cognitive spectra.

Assessing factor structures and measurement invariance can inform about the validity of CES-D and its short forms among older individuals. In addition, investigating the association of genetic risk of depression with different CES-D versions can provide further evidence for the utility of CES-D. Polygenic risk score of major depressive disorder (MDD-PRS), summing up the small effects of multiple single nucleotide polymorphisms (SNPs) with 8.7% SNP-based heritability, explains 1.9% of the genetic liability to major depressive disorder and is related to severity of depression [24]. It has been suggested that genetic factors related to depression vulnerability persist in aging [25], but this has been scarcely examined [26]. Inflammatory, vascular and neurodegenerative markers of accelerated biological aging may contribute to depression in late life [25], but these etiological factors may not be represented in MDD-PRS that is derived mainly from samples of middle-aged participants [24]. Thus, it needs to be examined whether the MDD-PRS is associated with CES-D in late old age.

The aim of this study was to compare factor structures of different versions of the CES-D and to test the measurement invariance regarding old-age groups and cognitive status by using a population-based sample including those in their 70's and those who are 90 years or older. We also investigated the association of MDD-PRS with different CES-D versions. We found that different versions of the CES-D showed measurement invariance for age and cognitive status, promoting the use of this instrument in studies of aging. Furthermore, MDD-PRS showed associations with all CES-D versions and most subscales.

## Materials and methods

### Participants

We used two sub-studies based on a population-based older Finnish Twin Cohort (FTC) study that includes Finnish twins from same-sex pairs born before 1958 [27]. A total of 1698 individuals (age M = 73.6±1.6, born 1938–1944) were from MEMTWIN II (participation rate of 67%) sub-study with data collected between August 29, 2013 to July 4, 2017 [28]. We split the MEMTWIN II sample by requiring distinct family numbers: the first co-twins of a pair or twins without a co-twin formed the exploratory sample (n = 1034) and the second co-twins from full pairs formed the confirmatory sample (n = 664).

The older subsample were individuals from the FTC who were over 90 years old or reached 90 years of age during the ongoing NONAGINTA study data collection that started June 1, 2020. We included the first 150 participants (reached on January 10, 2023) with a mean age of 91.4 (SD 1.9) years and participation rate of 28%. Of these, 134 had CES-D data (see S1 File for additional information about the samples).

MEMTWIN II study protocol was approved by the Ethics Committee of the Hospital District of Southwest Finland and NONAGINTA study protocol was approved by the Ethics Committee of the Hospital District of Helsinki and Uusimaa. For both sub-studies, written informed consents were obtained and the principles outlined in the declaration of Helsinki were followed.

### Measures

**Center for Epidemiological Studies Depression scale.** We used the Finnish version of the CES-D 20 and its shorter versions as measures of depressive symptoms [29]. The CES-D is a 20-item self-report instrument assessing depressive symptoms on a four-point scale (0–3), with a maximum score of 60 points (Table 1). Four items are reverse-coded (Good, Hopeful, Happy, Enjoyed). The CES-D has shown good validity and reliability in studies of older adults [12]. As all items of the CES-D were required for factor analyses, only participants with complete CES-D data were used in this study. Additionally, the scale was administered in pen-and-paper format, leading us to omit responses that fell between options (e.g., participant marked their choice between point 1 and 2) as equal number of response options are required for factor analyses.

**Modified Telephone Interview for Cognitive Status.** The modified Telephone Interview for Cognitive Status (TICS-m) was used as a measure of cognitive functioning [28], see S1 File for more details.

**Polygenic risk score of major depressive disorder.** Technical details of genotyping, imputation and polygenic risk score calculation have been described elsewhere [30]. MDD-PRS was derived using GWAS summary statistics of major depressive disorder [24] and used as a continuous predictor of CES-D scores. The total number of single nucleotide polymorphisms (SNP) used for polygenic score calculation was 1147810. The number of samples used for the original genome-wide association study was 173005.

**Table 1. Items of the CES-D 20 and their inclusion in published subscales.**

| Item | Name used in the study | Included in the 8-item version | Included in the 15-item version |
|---|---|---|---|
| 1. I was bothered by things that usually don't bother me | Bothered | | x |
| 2. I did not feel like eating; my appetite was poor | Appetite | | x |
| 3. I felt that I could not shake off the blues even with help from my family or friends | Blues | | x |
| 4. I felt that I was just as good as other people | Good | | |
| 5. I had trouble keeping my mind on what I was doing | Concentrating | | x |
| 6. I felt depressed | Depressed | x | x |
| 7. I felt that everything I did was an effort | Effort | x | x |
| 8. I felt hopeful about the future | Hopeful | | |
| 9. I thought my life had been a failure | Failure | | x |
| 10. I felt fearful | Fearful | | x |
| 11. My sleep was restless | Sleep | x | x |
| 12. I was happy | Happy | x | |
| 13. I talked less than usual | Talk | | x |
| 14. I felt lonely | Lonely | x | x |
| 15. People were unfriendly | Unfriendly | | |
| 16. I enjoyed life | Enjoyed | x | |
| 17. I had crying spells | Crying | | x |
| 18. I felt sad | Sad | x | x |
| 19. I felt that people dislike me | Dislike | | x |
| 20. I could not get going | Going | x | x |

This study uses the Finnish translation of the CES-D 20 and the items included in the table are from the original study by Radloff (1977). Items 4, 8, 12, and 16 are reverse-coded.

## Statistical analyses

The exploratory sample (n = 1034) from MEMTWIN II was used for exploratory factor analyses (EFA) of the CES-D 20 and the CES-D 8. EFAs were ran first instead of trying to confirm previous factor structures because 1) we used the Finnish version of the CES-D and different language versions have yielded different factor structures for both short and long forms, 2) many previous factor analyses have used analytic approaches which might not be ideal for ordinal indicators, leading to possible biases in factor structures [31], and 3) the split samples still exceeded common recommendations for sample sizes in factor analysis [32]. Fig 1 shows which analyses were conducted in each subsample of MEMTWIN II.

Confirmatory factor analyses (CFA) were run in the confirmatory MEMTWIN II sample (n = 664) based on the factor structures identified in EFAs for both CES-D 8 and CES-D 20. We were also interested to see whether we could confirm two structures proposed in the earlier literature. First, we considered the 15-item unidimensional structure reported in a sample of young individuals from the US [10]. In their study, the authors suggested that the positively worded four items in the CES-D 20 are not necessary and the items Unfriendly and Dislike were interchangeable (with the item Unfriendly dropped from the 15-item version). Furthermore, the researchers used polychoric correlations and an estimation method suitable for ordinal data, lending their results suitable for replication. We considered the 15-item structure worth investigating as it is more comprehensive than the CES-D 8, but unidimensional unlike the CES-D 20.

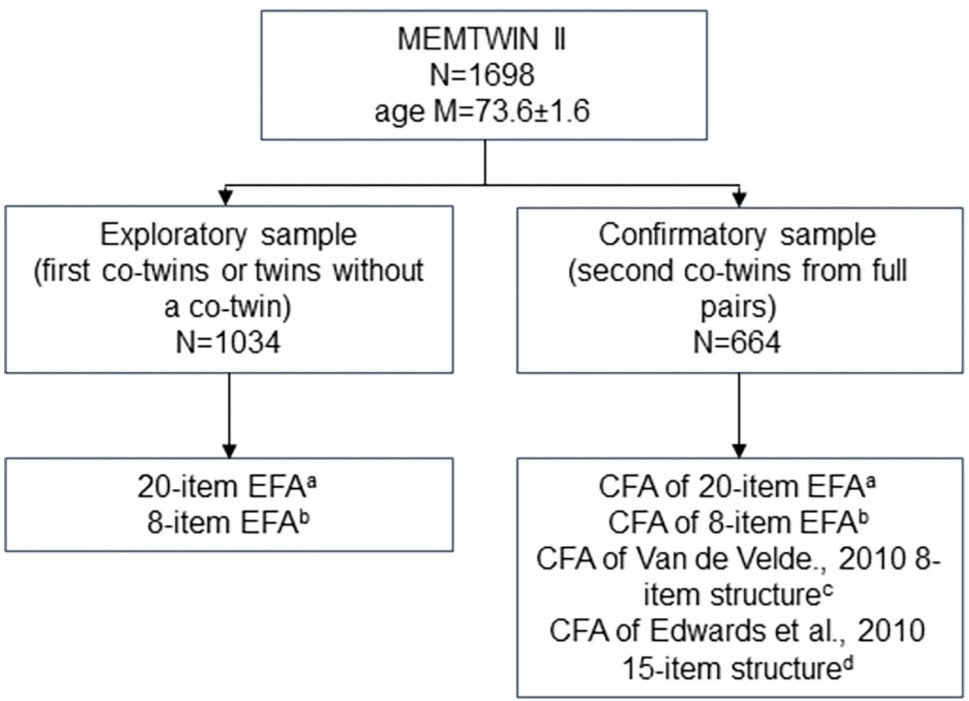

**Fig 1. Exploratory and confirmatory factor analyses.** First, exploratory factor analyses were conducted for all 20 items of the CES-D and for the 8-item subset. Then, these structures were tested in confirmatory factor analysis in addition to two theory-driven models by Van de Velde et al. (2010) and Edwards et al. (2010). CFA = confirmatory factor analysis; EFA = exploratory factor analysis.

Next, owing to the challenges posed by the positively worded items also in the CES-D 8 items, we considered a unidimensional CES-D 8 structure with added covariances between Happy & Enjoy, which has resulted in good fits in large studies [11, 16]. The CES-D 20 and the subscales under investigation in this study are shown in Table 1. Additional information about analyses examining suitability of the data for factor analyses, model estimation and reporting are in S1 File.

Measurement invariance testing was performed for age group comparisons (70-year-olds from the MEMTWIN II vs 90-year-olds from the NONAGINTA) and cognitive status (impaired versus unimpaired in MEMTWIN II). First, we estimated configural invariance (M1), where only the number and pattern of factor loadings are equal. Next, invariance for thresholds was investigated (M2), followed by invariance for thresholds and factor loadings (M3, metric invariance) and thresholds, factor loadings and intercepts (M4, scalar invariance). These four invariance models were tested with the CES-D 20 and CES-D 8 models derived from our exploratory analyses and the two theory-driven models–CES-D 15 [10] and CES-D 8 [11] with correlated residuals–mentioned previously (structures from Fig 1 confirmatory analyses). For comparison of latent means across groups, scalar invariance should be met [33]. Measurement invariance testing was done following Wu & Estabrook [34] using *semTools* for generating appropriate *lavaan* syntax for model specification. Theta parameterization was used in the models.

Differences between age groups and in cognitively impaired versus cognitively healthy were compared with regression analyses. MDD-PRS associations with CES-D versions were studied with linear regression analysis adjusting for age, sex and population stratification (10 genetic principal components). Clustered standard errors were used to account for non-independence of observations (twins within families, which were the primary sampling units).

All analyses were conducted with R version 4.2.2 [35]. R packages *psych* [36], *lavaan* [37], and *semTools* [38] were used for factor analyses and measurement invariance testing. Polychoric correlations were used in model estimation and scaled fit indices are reported. Regression analyses with clustered standard errors were performed using *estimatr* [39], and adjustments for multiple comparisons in these analyses were made following Holm [40]. All R code is available at https://osf.io/kxzta/.

## Results

### Sample characteristics

Demographic and clinical characteristics of the samples are described in Table 2. A total of 260 (18.7%) MEMTWIN II participants were classified as cognitively impaired based on TICS-m cut-offs. The missing data distributions of CES-D items did not differ between cognitively unimpaired and cognitively impaired groups ($\chi^2$ [19] = 7.43, p = 0.99) or between the MEMTWIN II and NONAGINTA cohorts ($\chi^2$ [19] = 6.43, p = 0.99). Additional results concerning response style and missing values are presented in S2 File.

### Exploratory factor analyses

A total of 1034 individuals (first member of a twin pair and all twins without a co-twin in the study) comprised the exploratory sample and these data were used for EFA. Prior to EFA, Kaiser-Meyer-Olkin coefficient of sampling adequacy was examined and found adequate for the 20 items and 8 items (.93 and .85, respectively). Bartlett's test of sphericity was found highly significant for both item pools ($p < .01 \times 10^{-70}$). Polychoric parallel analyses were ran using the *psych* package and 50 iterations; the results suggested four to five factors for the full scale and three factors for the 8-item subscale.

A four-factor solution in the EFA for the CES-D 20 (factors explaining 11–20% of the common variance) was considered the best, although this solution still had issues with fit (Root Mean Square Error of Approximation [RMSEA] = 0.084 [90% CI 0.079–0.089], Tucker-Lewis Index [TLI] = 0.901). The 20 items loaded largely onto the same factors as in the original Radloff [9] study with loadings ranging .45–.75. The only exceptions were Bothered and Sleep, which now loaded onto a depressed affect factor rather than to a factor with Appetite, Effort and Going. Of note, Dislike and Unfriendly loaded on their own factor, and all positively worded items and Failure loaded on their own factor. A five-factor solution yielded a negligible improvement in fit, and the fifth factor explained under 10% of the variance. Weighing these

**Table 2. Clinical and demographic characteristics of the study sample.**

|  | Total MEMTWIN II | Exploratory MEMTWIN II | Confirmatory MEMTWIN II | NONAGINTA | Difference, β (95% CI) |
|---|---|---|---|---|---|
| n | 1698 | 1034 | 664 | 134 |  |
| Age, M (SD) | 73.6 (1.6) | 73.9 (1.7) | 73.3 (1.4) | 91.4 (1.9) |  |
| Men, n (%) | 847 (49.9%) | 528 (51.1%) | 319 (48%) | 48 (35.8%) |  |
| TICS-m, M (SD) | 33.5 (5.1) | 33.4 (5.1) | 33.8 (5) | NA |  |
| CES-D 20, M (SD) | 7.7 (7.4) | 8.2 (7.7) | 6.9 (6.9) | 14.6 (9.4) | 2.67 (1.97–3.37) |
| CES-D 15, M (SD) | 4.8 (5.4) | 5.2 (5.6) | 4.2 (4.8) | 9.0 (7.5) | 1.87 (1.27–2.48) |
| CES-D 8, M (SD) | 3.8 (3.7) | 4.1 (3.8) | 3.5 (3.5) | 7.0 (4.8) | 1.15 (0.82–1.47) |

Difference is standardized regression coefficient for the grouping variable between cohorts (reference = MEMTWIN II), with CES-D scores as dependent variables, sex and group as independent variables and clustered standard errors for twin structures. NA = not available.

findings against the increase in model complexity and deviation from models in the prior literature led us to consider the more parsimonious four-factor model as the better option.

For the 8-item model, 3 factors were extracted with a better fit (RMSEA = .068 [90% CI .049-.089], TLI = .97) and loadings .30–.67, the lowest loading being on the item Sleep. The three-factor CES-D 8 model had a similar structure to the CES-D 20 but with fewer items and neither item of the Unfriendly/Dislike factor. A pair of two-item factors emerged (Enjoyed & Happy and Effort & Going), explaining 20% and 19% of the common variance, respectively, with the final four-item factor (Sad, Depressed, Lonely, Sleep) explaining 30% of the variance. Items with positive content loaded onto their own factors in both CES-D 20 and CES-D 8.

### Confirmatory factor analyses

In the CFA, the two EFA-driven models and two theory-driven models generally fit well to the data (Table 3). A slight exception was found for the CES-D 8 unidimensional model, which had RMSEA = 0.078 (90% CI 0.063–0.093) and non-overlapping RMSEA confidence intervals compared to the other three models, but comparative fit indices and SRMR were also within commonly accepted boundaries in this model. The lowest reliability values were found for the 2-item Dislike & Unfriendly subscale ($\alpha$ = .63 and $\omega_T$ = .67); as expected with factor models with varying loadings, $\omega_T$ provided higher reliability estimates than $\alpha$. Factor loadings varied .55-.96 in CES-D 20, .53-.90 in CES-D 8 three-factor model, .53-.83 in CES-D 15 and .53-.81 in the unidimensional CES-D 8 (Table 4). Of note, the item Sleep had the lowest factor loading in all models. The correlations between factors were $\geq$ 0.55 in the multi-factor models (S1 Table).

### Measurement invariance for age and cognition

Measurement invariance for age was analyzed comparing 70- to 90-year-olds using MEM-TWIN II and NONAGINTA samples, respectively. Models with configural invariance, threshold invariance, threshold invariance with loading invariance, and threshold invariance with loading invariance and intercept invariance are presented in Table 5. The 1-factor CES-D 8 model showed initially only a mediocre fit that improved in more constrained models. All scalar invariance models showed a decrease in fit compared to the metric invariance models as assessed by the $\chi^2$-test, and all but the CES-D 8 3-factor model also showed a decrease in fit when comparing metric models to models with threshold invariance. However, considering the stability (or even improvement) of the other fit indices, we interpret the $\chi2$-test results insufficient to reject invariance in more constrained models. These results indicated scalar measurement invariance for age. The mean differences between older and younger old adults were significant for the CES-D 20 ($\beta$ = 2.67 [95% CI 1.97–3.37]), the CES-D 15 ($\beta$ = 1.87 [95% CI 1.27–2.48]) and the CES-D 8 ($\beta$ = 1.15, [95% CI .82–1.47] with over 90-year-olds having markedly more depressive symptoms (Table 2).

**Table 3. Summary of the confirmatory factor analysis results.**

| Scale | Factors, n | $\chi2$ | df | RMSEA | RMSEA 90% CI | SRMR | CFI | TLI |
|---|---|---|---|---|---|---|---|---|
| CES-D 20 | 4 | 434 | 164 | 0.050 | 0.044–0.056 | 0.063 | 0.966 | 0.960 |
| CES-D 8 | 3 | 32 | 17 | 0.036 | 0.016–0.055 | 0.029 | 0.996 | 0.993 |
| CES-D 15[a] | 1 | 238 | 90 | 0.049 | 0.042–0.057 | 0.054 | 0.972 | 0.967 |
| CES-D 8[b] | 1 | 98 | 19 | 0.078 | 0.063–0.093 | 0.053 | 0.977 | 0.967 |

CFI = Comparative Fit Index, RMSEA = Root Mean Square Error of Approximation, SRMR = Standardized Root Mean Residual, TLI = Tucker Lewis Index.

[a] Edwards et al., 2010

[b] Van de Velde et al., 2010

**Table 4. Summary of factor loadings and reliabilities in confirmatory factor analyses.**

| | CES-D 20 | | | | CES-D 8 | | | CES-D 15[a] | CES-D 8[b] |
|---|---|---|---|---|---|---|---|---|---|
| | Factor 1 | Factor 2 | Factor 3 | Factor 4 | Factor 1 | Factor 2 | Factor 3 | Factor 1 | Factor 1 |
| *Item* | | | | | | | | | |
| Crying | **0.663** | | | | | | | 0.671 | |
| Sad | **0.766** | | | | 0.744 | | | 0.750 | 0.729 |
| Depressed | **0.851** | | | | 0.834 | | | 0.829 | 0.811 |
| Lonely | **0.751** | | | | 0.762 | | | 0.698 | 0.744 |
| Blues | **0.802** | | | | | | | 0.797 | |
| Fearful | 0.717 | | | | | | | 0.700 | |
| Bothered | *0.643* | | | | | | | 0.643 | |
| Sleep | *0.552* | | | | 0.537 | | | 0.530 | 0.525 |
| Concentrating | | 0.696 | | | | | | 0.680 | |
| Appetite | | **0.668** | | | | | | 0.664 | |
| Effort | | **0.867** | | | | | 0.874 | 0.822 | 0.809 |
| Talk | | 0.670 | | | | | 0.887 | 0.624 | |
| Going | | **0.804** | | | | | | 0.748 | 0.797 |
| Good[c] | | | **0.695** | | | | | | |
| Hopeful[c] | | | **0.784** | | | | | | |
| Failure | | | 0.856 | | | | | 0.748 | |
| Happy[c] | | | **0.853** | | | 0.899 | | | 0.603 |
| Enjoyed[c] | | | **0.852** | | | 0.861 | | | 0.579 |
| Unfriendly | | | | **0.719** | | | | | |
| Dislike | | | | **0.960** | | | | 0.651 | |
| *Reliability* | | | | | | | | | |
| Cronbach's alpha (α) | 0.79 | 0.72 | 0.79 | 0.63 | 0.66 | 0.79 | 0.74 | 0.87 | 0.8 |
| Omega total ($\omega_T$) | 0.82 | 0.73 | 0.81 | 0.67 | 0.72 | 0.8 | 0.72 | 0.88 | 0.82 |

Note. For CES-D 20, items loading onto the same factor as in the original study by Radloff (1977) are bolded, those loading onto a different factor are italicized, and those in normal typeface did not load onto any principal component in Radloff (1977).

[a] Edwards et al., 2010

[b] Van de Velde et al., 2010

[c] Reverse-scored items

After excluding 48 individuals with incomplete data for calculating TICS-m total score, measurement invariance for cognition was analyzed in the remaining MEMTWIN II participants (unimpaired cognition n = 1390, impaired cognition n = 260; Table 5). In measurement invariance testing for cognition, all models fit well with the added equality constraints except for the unidimensional CES-D 8. For the 1-factor CES-D 8 model, RMSEA values in the configural model indicate a mediocre fit; the more constrained models show better RMSEA values, but the confidence interval in the scalar model still exceeds the cut-off for good fit. In all, scalar measurement invariance for cognition was supported by these analyses. No significant differences were found in CES-D 20, CES-D 15 or CES-D 8 total scores between cognitively unimpaired and cognitively impaired individuals (S2 Table).

## Associations with genetic risk for depression

Finally, we looked at the associations of genetic risk (MDD-PRS) to CES-D scores (Table 6). MDD-PRS was similarly associated with all three CES-D total scores (CES-D 20: β = 0.084 [95% CI 0.029–0.138]; CES-D 15: β = 0.071 [95% CI 0.017–0.126]; CES-D 8: β = 0.082 [95% CI

**Table 5. Summary of measurement invariance tests for age and cognitive groups.**

| Scale | Model | χ2 | df | χ2 difference p | CFI | Δ CFI | TLI | RMSEA (90% CI) | SRMR |
|---|---|---|---|---|---|---|---|---|---|
| *For age* | | | | | | | | | |
| CES-D 20 (4 factors) | M1 | 859 | 328 | - | 0.968 | - | 0.963 | 0.050 (0.047–0.053) | 0.058 |
| | M2 | 862 | 347 | 0.97 | 0.969 | +0.000 | 0.966 | 0.048 (0.045–0.052) | 0.058 |
| | M3 | 904 | 363 | 0.01 | 0.971 | +0.002 | 0.969 | 0.046 (0.043–0.049) | 0.058 |
| | M4 | 975 | 379 | < 0.001 | 0.972 | +0.001 | 0.972 | 0.044 (0.041–0.047) | 0.059 |
| CES-D 8 (3 factors) | M1 | 42 | 34 | - | 0.995 | - | 0.991 | 0.041 (0.030–0.052) | 0.026 |
| | M2 | 44 | 42 | 0.79 | 0.995 | +0.000 | 0.993 | 0.036 (0.025–0.046) | 0.026 |
| | M3 | 49 | 47 | 0.37 | 0.996 | +0.001 | 0.995 | 0.031 (0.021–0.041) | 0.027 |
| | M4 | 61 | 52 | 0.02 | 0.995 | -0.001 | 0.994 | 0.033 (0.023–0.042) | 0.027 |
| CES-D 15 (1 factor) | M1 | 435 | 180 | - | 0.97 | - | 0.965 | 0.053 (0.048–0.057) | 0.05 |
| | M2 | 437 | 194 | 0.97 | 0.971 | +0.001 | 0.968 | 0.050 (0.046–0.055) | 0.05 |
| | M3 | 462 | 208 | 0.11 | 0.974 | +0.003 | 0.974 | 0.046 (0.042–0.050) | 0.05 |
| | M4 | 514 | 222 | < 0.001 | 0.976 | +0.002 | 0.977 | 0.043 (0.039–0.047) | 0.05 |
| CES-D 8 (1 factor) | M1 | 160 | 38 | - | 0.976 | - | 0.964 | 0.083 (0.073–0.092) | 0.049 |
| | M2 | 162 | 46 | 0.79 | 0.976 | 0.000 | 0.97 | 0.076 (0.067–0.084) | 0.049 |
| | M3 | 185 | 53 | 0.01 | 0.979 | +0.003 | 0.977 | 0.066 (0.058–0.074) | 0.05 |
| | M4 | 229 | 60 | < 0.001 | 0.976 | +0.002 | 0.978 | 0.065 (0.058–0.073) | 0.05 |
| *For cognition* | | | | | | | | | |
| CES-D 20 (4 factors) | M1 | 790 | 328 | - | 0.97 | - | 0.966 | 0.05 (0.046–0.053) | 0.059 |
| | M2 | 799 | 346 | 0.45 | 0.971 | +0.001 | 0.968 | 0.048 (0.045–0.052) | 0.059 |
| | M3 | 807 | 362 | 0.93 | 0.973 | +0.002 | 0.972 | 0.045 (0.042–0.048) | 0.059 |
| | M4 | 822 | 378 | 0.84 | 0.977 | +0.004 | 0.977 | 0.041 (0.037–0.044) | 0.06 |
| CES-D 8 (3 factors) | M1 | 38 | 34 | - | 0.995 | - | 0.993 | 0.038 (0.026–0.50) | 0.027 |
| | M2 | 41 | 42 | 0.58 | 0.996 | 0.000 | 0.994 | 0.034 (0.022–0.045) | 0.027 |
| | M3 | 41 | 47 | 0.98 | 0.997 | +0.002 | 0.997 | 0.025 (0.012–0.037) | 0.027 |
| | M4 | 45 | 52 | 0.35 | 0.997 | +0.000 | 0.997 | 0.023 (0.010–0.034) | 0.027 |
| CES-D 15 (1 factor) | M1 | 406 | 180 | - | 0.973 | - | 0.969 | 0.052 (0.048–0.057) | 0.052 |
| | M2 | 414 | 193 | 0.34 | 0.974 | +0.001 | 0.972 | 0.050 (0.045–0.054) | 0.052 |
| | M3 | 421 | 207 | 0.95 | 0.978 | +0.004 | 0.978 | 0.044 (0.040–0.049) | 0.052 |
| | M4 | 439 | 221 | 0.56 | 0.982 | +0.004 | 0.982 | 0.039 (0.034–0.043) | 0.052 |
| CES-D 8 (1 factor) | M1 | 155 | 38 | - | 0.976 | - | 0.965 | 0.082 (0.073–0.092) | 0.05 |
| | M2 | 158 | 46 | 0.58 | 0.976 | 0.000 | 0.971 | 0.076 (0.067–0.085) | 0.05 |
| | M3 | 160 | 53 | 0.95 | 0.981 | +0.005 | 0.98 | 0.062 (0.054–0.071) | 0.051 |
| | M4 | 169 | 60 | 0.35 | 0.984 | +0.003 | 0.985 | 0.054 (0.046–0.062) | 0.051 |

Models: M1 = configural invariance (equal number and pattern of factor loadings); M2 = threshold invariance (equal thresholds); M3 = metric invariance (equal thresholds and factor loadings); M4 = scalar invariance (equal thresholds, factor loadings and intercepts). Model comparisons are sequential: M1 vs M2; M2 vs M3; M3 vs M4. CFI = Comparative Fit Index, RMSEA = Root Mean Square Error of Approximation, SRMR = Standardized Root Mean Residual, TLI = Tucker Lewis Index, Δ CFI = change in Comparative Fit Index (M2—M1; M3—M2; M4—M3)

0.027–0.137]). The analyses were adjusted for age, sex and 10 first genetic principal components for population stratification, the clustering of individuals as twin siblings was taken into account and the analyses were corrected for multiple comparisons [40]. The associations between MDD-PRS and CES-D 20 and CES-D 8 subscale scores varied from β = .075 to β = .081 in significant associations. In contrast, MDD-PRS did not have statistically significant associations to CES-D 20 subscale 2 (Concentrating, Appetite, Effort, Talk, Going), subscale 4 (Unfriendly, Dislike) or CES-D 8 subscale 3 (Effort, Going). These associations were also

**Table 6. Associations between polygenic risk scores for depression and CES-D scores (n = 1529).**

| Scale | β (95% CI) | SE | p | Adjusted p | Adjusted R² |
|---|---|---|---|---:|---|
| CES-D 20 Total Score | 0.084 (0.029–0.138) | 0.028 | 0.003 | **0.029** | 0.036 |
| CES-D 15 Total Score | 0.071 (0.017–0.126) | 0.028 | 0.010 | **0.041** | 0.035 |
| CES-D 8 Total Score | 0.082 (0.027–0.137) | 0.028 | 0.004 | **0.033** | 0.029 |
| CES-D 20 Subscale 1 | 0.079 (0.025–0.134) | 0.028 | 0.004 | **0.033** | 0.042 |
| CES-D 20 Subscale 2 | 0.045 (-0.009–0.099) | 0.028 | 0.104 | 0.312 | 0.018 |
| CES-D 20 Subscale 3 | 0.080 (0.025–0.134) | 0.028 | 0.004 | **0.033** | 0.026 |
| CES-D 20 Subscale 4 | 0.040 (-0.012–0.092) | 0.027 | 0.135 | 0.312 | 0.018 |
| CES-D 8 Subscale 1 | 0.081 (0.026–0.135) | 0.028 | 0.004 | **0.033** | 0.035 |
| CES-D 8 Subscale 2 | 0.075 (0.021–0.129) | 0.027 | 0.006 | **0.033** | 0.016 |
| CES-D 8 Subscale 3 | 0.029 (-0.025–0.083) | 0.027 | 0.286 | 0.312 | 0.016 |

Subscales are the unit-weighted sum scores of the items comprising the respective factors in Table 4. Models included polygenic scores, 10 principal components for population stratification, age, sex and family number as a cluster to account for the non-independence of standard errors in twins. β = standardized regression coefficient, SE = standard error. P-values in multiple comparisons are adjusted following Holm.

smaller in magnitude (β = .029 to β = .045). In MDD-PRS decile plots (Fig 2), the differences between different deciles on CES-D scores were quite low, but CES-D scores were the highest in the top decile for MDD-PRS.

## Discussion

We found psychometric support for the shorter and longer forms of the CES-D in older adults. Our analyses suggest that the CES-D measures the same constructs in the presence or absence

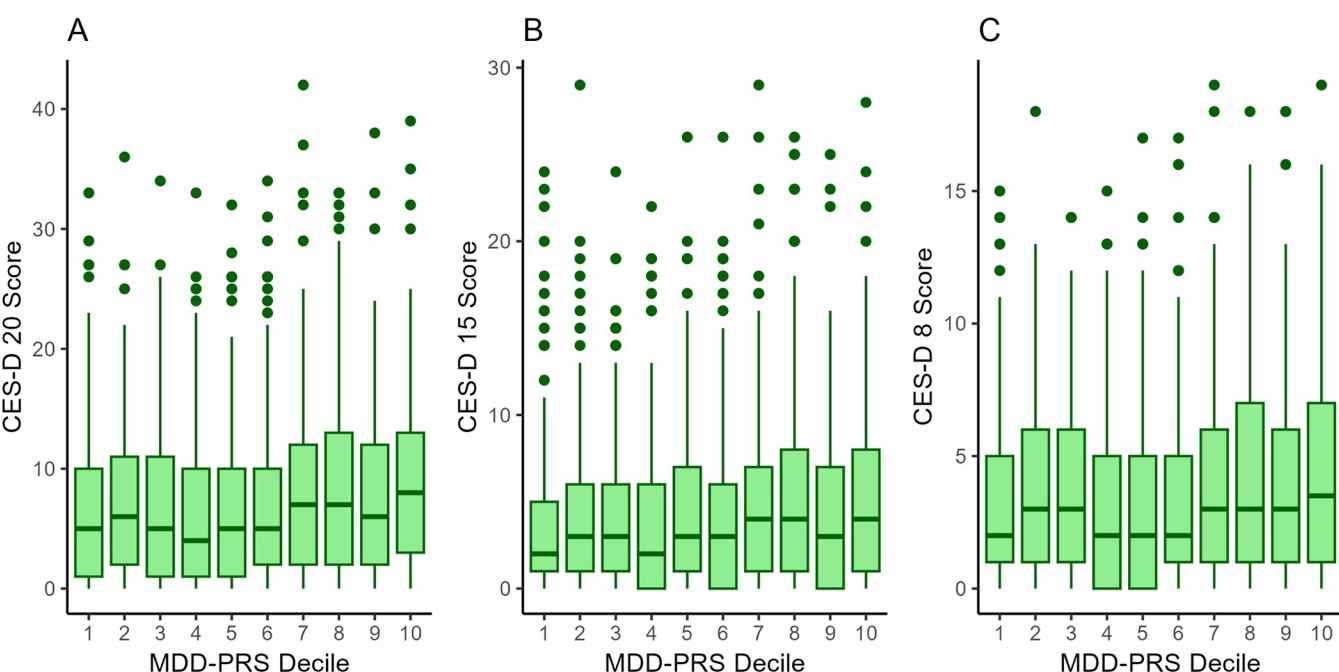

**Fig 2. Box-and-whiskers plot of CES-D scores by MDD-PRS deciles.** (A) CES-D 20. (B) CES-D 15. (C) CES-D 8. Boxes show lower quartile, median and upper quartiles, whiskers indicate a maximum of 1.5 times interquartile range from the corresponding upper and lower quartiles. CES-D = Center for Epidemiological Studies depression scale; MDD-PRS = polygenic risk score of major depressive disorder.

of cognitive impairment and in over 90-year-olds and those in their 70's. Over 90-year-olds had substantially more depressive symptoms than individuals in early old age and higher polygenic risk score associated with higher CES-D scores.

The CES-D 20 has some limitations as some items may load onto a factor based on shared variance unrelated to depression (e.g. wording of the items, overlap in item content in Unfriendly and Dislike) [19]. We found the 15-item unidimensional structure [10] without the positive items and the redundant Unfriendly item to have a good fit to the data and the scale showed measurement invariance for cognition and age. Using the 15-item version as a unidimensional measure of depressive symptoms when the study participation is not overly burdensome seems reasonable.

The 3-factor CES-D model showed a better fit than the unidimensional model for 8-item version [11], comparable fit to the CES-D 15 and the correlations among the three factors were ≥ 0.55. Both CES-D 8 models showed measurement invariance for cognition and age and the total score of CES-D 8 showed similar magnitude of association with MDD-PRS as the full-length CES-D. Overall, our results support using both, the 3-factor and the unidimensional CES-D 8 in older adults, particularly when the aim is to minimize the burden of administering a depression rating scale. Thus, for studies of older individuals, the CES-D 8 seems adequate for measuring depressive symptoms even in late old age and in those with cognitive impairment.

Our results showing measurement invariance for age, together with the previously demonstrated temporal measurement invariance [15], indicate that the CES-D can be used to compare depressive symptoms across age strata in older adults. Our study also adds to the literature on the self-assessment of depressive symptoms in individuals with cognitive impairment. Due to anosognosia and cognitive impairment, informant- or clinician-ratings of depression are used with individuals with cognitive decline. However, informant-rated symptoms correlate poorly with those of self-report [41]. Importantly, the CES-D items seem invariant for cognitive status [21] and the diagnostic accuracy seems to be similar in individuals with mild dementia and cognitively unimpaired older adults [20]. A recent study also suggests that the discordance between self-reported and informant-reported depressive symptoms is not related to cognitive impairment [42]. Using self-reported depressive symptoms may provide data that could not be accessed with informant ratings, and it appears that these data are as meaningful as they are in older adults without cognitive impairment.

In our study, over 90-year-olds had almost two-fold scores in CES-D compared to those in their 70's: this does not necessarily mean that depressive symptoms increase from old age to very old age, but can reflect cohort differences whereby younger cohorts experience less depressive symptoms than older cohorts at the same age over 20 years ago [43]. The data collection in NONAGINTA coincided with the COVID-19 pandemic, possibly influencing the CES-D scores in 90-year-olds.

Finally, the associations of MDD-PRS with CES-D scores were of similar magnitude in different versions. An earlier study with two-factor CES-D 8 found associations between MDD-PRS and both factors [26]. In contrast, we found that the factor comprising items "Going" and "Effort" had no association with MDD-PRS. This lack of association may relate to these items capturing aspects of apathy and executive dysfunction that are not specific to depression.

The CFA methods used here are appropriate for ordinal data, but as a limitation we note that they may yield higher values than maximum likelihood-based approaches. Improvement of fit indexes with added equality constraints for age and cognition is somewhat surprising, but has been reported in previous depression scale measurement invariance literature using ordinal data [44]. This observation may relate to insensitivity of some fit indexes, especially

with high degrees of freedom [45]. Furthermore, individuals with severe dementia likely did not participate in our study, limiting the range of conclusions about the invariance for cognitive status. Previous research indicates that cognitive decline begins to affect self-reported depressive symptoms at the moderate stage of dementia [46]. It is also possible that anosognosia could have affected the scores of individuals with cognitive impairment. Therefore, not adjusting for anosognosia can be considered a limitation of the study.

The strength of our study was using population-based samples and comparisons between cohorts from the same underlying population including representants also from the oldest-old age group, comprising the fasted growing age group in the world [47]. Furthermore, our sample size allowed us to examine both data-driven and theory-driven factor structures of the CES-D. The measurement invariance analyses in this study use the relatively recent advances in conducting these analyses using ordinal data [34]. We also used a validated measure of cognitive performance with an education adjustment to reduce the number of false positive cases of cognitive impairment [28].

## Conclusion

In conclusion, the CES-D and its short forms can be used in older adults with cognitive impairment or in 90-year-olds. The polygenic risk for depression is captured by the CES-D in older adults.

## Supporting information

**S1 File. Additional information about participants and methods.**
(DOCX)

**S2 File. Missing values and response style.**
(DOCX)

**S1 Table. Correlations among factors in multi-factor models.**
(DOCX)

**S2 Table. CES-D score differences between those with cognitive impairment and those with normal cognition in the MEMTWIN II cohort.**
(DOCX)

## Acknowledgments

We thank the participants of the older Finnish Twin Cohort study. We thank Noora Lindgren, Kristiina Saanakorpi and Ulla Kulmala-Gråhn for data collection in the MEMTWIN II study and Mia Urjansson for data collection in the NONAGINTA study.

## Author Contributions

**Conceptualization:** Toni T. Saari, Maarit Piirtola.

**Data curation:** Teemu Palviainen, Anni Varjonen.

**Formal analysis:** Toni T. Saari.

**Funding acquisition:** Juha O. Rinne, Jaakko Kaprio, Eero Vuoksimaa.

**Investigation:** Juha O. Rinne, Jaakko Kaprio, Eero Vuoksimaa.

**Writing – original draft:** Toni T. Saari, Eero Vuoksimaa.

**Writing – review & editing:** Toni T. Saari, Maarit Piirtola, Aino Aaltonen, Teemu Palviainen, Anni Varjonen, Valtteri Julkunen, Juha O. Rinne, Jaakko Kaprio, Eero Vuoksimaa.

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
