## [Decision Letter · Decision Letter 0]

20 Aug 2024

PONE-D-24-14819Measurement invariance of the Center for Epidemiological Studies-Depression scale and associations with genetic risk in older adultsPLOS ONE

Dear Dr. Saari,

Thank you for submitting your manuscript to PLOS ONE. After careful consideration, we feel that it has merit but does not fully meet PLOS ONE’s publication criteria as it currently stands. Therefore, we invite you to submit a revised version of the manuscript that addresses the points raised during the review process.

I agree with both reviewers about the need for minor revisions. Please submit your revised manuscript by Oct 04 2024 11:59PM. If you will need more time than this to complete your revisions, please reply to this message or contact the journal office at plosone@plos.org. Please include the following items when submitting your revised manuscript:A rebuttal letter that responds to each point raised by the academic editor and reviewer(s). You should upload this letter as a separate file labeled 'Response to Reviewers'.A marked-up copy of your manuscript that highlights changes made to the original version. You should upload this as a separate file labeled 'Revised Manuscript with Track Changes'.An unmarked version of your revised paper without tracked changes. You should upload this as a separate file labeled 'Manuscript'.If applicable, we recommend that you deposit your laboratory protocols in protocols.io to enhance the reproducibility of your results. Protocols.io assigns your protocol its own identifier (DOI) so that it can be cited independently in the future. For instructions see: https://journals.plos.org/plosone/s/submission-guidelines#loc-laboratory-protocols. Additionally, PLOS ONE offers an option for publishing peer-reviewed Lab Protocol articles, which describe protocols hosted on protocols.io. Read more information on sharing protocols at https://plos.org/protocols?utm_medium=editorial-email&utm_source=authorletters&utm_campaign=protocols.

We look forward to receiving your revised manuscript.

Kind regards,

Diego A. Forero, MD; PhD

Academic Editor

PLOS ONE

Journal Requirements:

"Data collection and analyses in the Finnish Twin Cohort have been supported by the Academy of Finland (grants to JK: 312073, 336823, 352792). MEMTWIN II data collection was supported by the Sigrid Juselius foundation, The Academy of Finland (grants 133193 and 310962) and Finnish Governmental Research Funding (VTR). NONAGINTA study was funded by the Academy of Finland grants (320109 and 345988 to EV). EV was supported by the Academy Research Fellow grant 314639 and the Sigrid Jusélius Foundation. Open access funded by Helsinki University Library."

"JK has received support from Sigrid Jusélius Foundation, the European Union and the National Institutes of Health. JOR has consulted for Clinical Research Services Turku and acted as a member on the data monitoring committee for Lundbeck."

Reviewers' comments:

Reviewer's Responses to Questions

**Comments to the Author**

1. Is the manuscript technically sound, and do the data support the conclusions?

Reviewer #1: Yes

Reviewer #2: Yes

2. Has the statistical analysis been performed appropriately and rigorously? 

Reviewer #1: I Don't Know

Reviewer #2: Yes

3. Have the authors made all data underlying the findings in their manuscript fully available?

Reviewer #1: Yes

Reviewer #2: No

4. Is the manuscript presented in an intelligible fashion and written in standard English?

Reviewer #1: Yes

Reviewer #2: Yes

5. Review Comments to the Author

Reviewer #1: The manuscript has a logical and sound structure and methodological strengths, and it is written in standard, correct, and clear English.

As a suggestion, the statistical analyses should be confirmed.

Reviewer #2: To improve the clarity of the materials and methods section, it would be appropriate to add a figure of the participants in the study and to subdivide the measures section into subsections to improve its understanding by the reader to explain the instruments in more detail.Additionally, it is necessary to review the writing of the statistical analysis section to make it clearer to the reader.

6. PLOS authors have the option to publish the peer review history of their article (what does this mean?). If published, this will include your full peer review and any attached files.

Reviewer #1: **Yes: **Sandra Milena Molano Pacheco

Reviewer #2: **Yes: **Luz Angela Rojas-Bernal

---

## [Author Response · Author response to Decision Letter 0]

26 Aug 2024

(Please see attached response letter for fully formatted version)

We thank the reviewers and the editor for their encouraging and insightful comments which have improved the manuscript. The changes in the manuscript are noted by Tracked Changes and by page (P#) in italicized text in this response letter. The responses are listed in the following order: comments by the editor, comments by the reviewers in a separate document, and comments by the reviewers included in the manuscript decision message. 

Editor’s Comments

Response: We have ensured that our manuscript meets the journal’s style requirements. 

"Data collection and analyses in the Finnish Twin Cohort have been supported by the Academy of Finland (grants to JK: 312073, 336823, 352792). MEMTWIN II data collection was supported by the Sigrid Juselius foundation, The Academy of Finland (grants 133193 and 310962) and Finnish Governmental Research Funding (VTR). NONAGINTA study was funded by the Academy of Finland grants (320109 and 345988 to EV). EV was supported by the Academy Research Fellow grant 314639 and the Sigrid Jusélius Foundation. Open access funded by Helsinki University Library."

Response: Funders had no role in the study and the corresponding sentence has been added to the financial disclosure. 

"JK has received support from Sigrid Jusélius Foundation, the European Union and the National Institutes of Health. JOR has consulted for Clinical Research Services Turku and acted as a member on the data monitoring committee for Lundbeck."

Response: The updated Competing Interests statement has been added to the cover letter as requested: “This does not alter our adherence to PLOS ONE policies on sharing data and materials.”. 

Response: We have reviewed the reference list for completeness and correctness. None of the cited papers have been retracted. 

Reviewer comments

Peer Review “Measurement invariance of the Center for Epidemiological Studies-Depression scale and associations with genetic risk in older adults”

The manuscript aims to propose that different versions of the CES-D are capable of measuring depressive symptoms, particularly in individuals over the age of 90 and/or those with cognitive impairment. Higher MDD-PRS (Major Depressive Disorder Polygenic Risk Score) was associated with more depressive symptoms across various CES-D versions. 

Moreover, the authors suggest that the CES-D maintains consistent constructs in the presence or absence of cognitive impairment and among individuals over 90 years old as well as those in their 70s, even with a higher polygenic risk score associated with elevated CES-D scores.

Strengths:

- The exploratory and confirmatory factor analyses are thorough and consistent; the information and supporting results are clear.

- The results are presented in a clear and didactic manner.

Major Issues

1. The results related to the association between genetic risk for depression and CES – D scores, one of the aims of the study, could be explore further

Response: We attempted to be exhaustive in our analyses by looking at all CES-D scales and subscales in the regression analyses with MDD-PRS. The only addition we could think of was to examine the CES-D 20, 15 and 8 scores for each MDD-PRS decile. Thus, we added Figure 2 to show these distributions. 

P15: “In MDD-PRS decile plots (Fig2), the differences between different deciles on CES-D scores were quite low, but CES-D scores were the highest in the top decile for MDD-PRS.”

Minor issues

2. It’s no clear how in this study were controlled strange variables, such as anosognosia effect in self report

Response: Anosognosia can indeed affect the validity of self-report measures, including CES-D. We have added the following text to Discussion:

P17: “It is also possible that anosognosia could have affected the scores of individuals with cognitive impairment. Therefore, not adjusting for anosognosia can be considered a limitation of the study.”

3. The authors don’t include graphs particularly in the Results section (for example, these could be usefull in exploratory and confirmatory factor analyses sumarize)

Response: We agree with the reviewer that the results rely heavily on tables and text. We believe that the presentation of factor analyses as tables follows the conventional reporting of these analyses, especially with a considerable number of models (exploratory and confirmatory, measurement invariance) and is appropriate as such. However, we do agree with the reviewer that visualization can be helpful and have made a few additions to the manuscript. First, we added Fig1 to show how the MEMTWIN II sample was divided for exploratory and confirmatory analyses. Second, to display more information about the relationship between MDD-PRS and CES-D, we have added a panel of decile plots showing the CES-D scores for each decile of the MDD-PRS (Fig2). 

4. Explore further the associations with genetic risk for depression in the results.

Response: Please see our response to the comment 1. 

Additional Reviewer Comments:

5. Reviewer #1: The manuscript has a logical and sound structure and methodological strengths, and it is written in standard, correct, and clear English.

As a suggestion, the statistical analyses should be confirmed.

Response: We apologize, but it is not entirely clear to us what the referee means by “confirming statistical analyses” here. Please note, that we have provided code for the analyses which can be modified to be used in other data sets for similar data. 

6. Reviewer #2: To improve the clarity of the materials and methods section, it would be appropriate to add a figure of the participants in the study and to subdivide the measures section into subsections to improve its understanding by the reader to explain the instruments in more detail. Additionally, it is necessary to review the writing of the statistical analysis section to make it clearer to the reader.

Response: We have now divided the measures section into three subdivisions, namely CES-D, TICS-m and polygenic risk score. The full description of TICS-m is lengthy, which is why the detailed description of the measure is shown in the Supplementary File 1. Figure 1 has been added to show how the models were ran in the subsamples of MEMTWIN II. 

P7: “Fig1 shows which analyses were conducted in each subsample of MEMTWIN II.”

This Figure is now also referenced in the paragraph concerning measurement invariance testing. We hope that this figure helps the reader in conceptualizing which models were used in the measurement invariance part of the paper. 

P8-9: “These four invariance models were tested with the CES-D 20 and CES-D 8 models derived from our exploratory analyses and the two theory-driven models – CES-D 15 [10] and CES-D 8 [11] with correlated residuals – mentioned previously (structures from Fig1 confirmatory analyses).”

In the manuscript, we have tried to be as precise as possible about the modelling decisions and added supplementary text to complement the main text. We did also provide a link to the R code for running the analyses, which can be checked by the reader of this paper as verbal descriptions of analyses are often ambiguous. Despite these efforts, we understand that some parts of the text can still be unclear. We are happy to clarify any specific parts of the statistical analyses that the reviewer considers particularly problematic to follow.

---

## [Decision Letter · Decision Letter 1]

3 Oct 2024

Measurement invariance of the Center for Epidemiological Studies-Depression scale and associations with genetic risk in older adults

PONE-D-24-14819R1

Dear Dr. Saari,

We’re pleased to inform you that your manuscript has been judged scientifically suitable for publication and will be formally accepted for publication once it meets all outstanding technical requirements.

Kind regards,

Diego A. Forero, MD; PhD

Academic Editor

PLOS ONE

Additional Editor Comments (optional):

Reviewers' comments:

Reviewer's Responses to Questions

**Comments to the Author**

1. If the authors have adequately addressed your comments raised in a previous round of review and you feel that this manuscript is now acceptable for publication, you may indicate that here to bypass the “Comments to the Author” section, enter your conflict of interest statement in the “Confidential to Editor” section, and submit your "Accept" recommendation.

Reviewer #1: All comments have been addressed

Reviewer #2: All comments have been addressed

2. Is the manuscript technically sound, and do the data support the conclusions?

Reviewer #1: Yes

Reviewer #2: (No Response)

3. Has the statistical analysis been performed appropriately and rigorously? 

Reviewer #1: Yes

Reviewer #2: (No Response)

4. Have the authors made all data underlying the findings in their manuscript fully available?

Reviewer #1: Yes

Reviewer #2: (No Response)

5. Is the manuscript presented in an intelligible fashion and written in standard English?

Reviewer #1: Yes

Reviewer #2: (No Response)

6. Review Comments to the Author

Reviewer #1: Strengths:

• The authors have addressed the editor’s and reviewers' comments effectively, which has resolved previous concerns.

• Figures 1 and 2 are valuable aids in enhancing the clarity of the presented results, making it easier for readers to engage with the article.

• The discussion appropriately includes limitations, such as the potential impact of anosognosia on the validity of self-report measures.

Major Issues:

• No major issues were identified in the latest version of the manuscript.

Suggestions:

• It is recommended that the authors consider incorporating strategies to mitigate the impact of anosognosia in future research.

Reviewer #2: (No Response)

7. PLOS authors have the option to publish the peer review history of their article (what does this mean?). If published, this will include your full peer review and any attached files.

Reviewer #1: No

Reviewer #2: **Yes: **Luz Angela Rojas-Bernal

---

## [Editor Report · Acceptance letter]

17 Oct 2024

PONE-D-24-14819R1 

PLOS ONE

Dear Dr. Saari, 

I'm pleased to inform you that your manuscript has been deemed suitable for publication in PLOS ONE. Congratulations! Your manuscript is now being handed over to our production team.

Kind regards, 

on behalf of

Dr. Diego A. Forero 

Academic Editor

PLOS ONE